# Pilot Study: Does Contamination with Enniatin B and Beauvericin Affect the Antioxidant Capacity of Cereals Commonly Used in Animal Feeding?

**DOI:** 10.3390/plants10091835

**Published:** 2021-09-03

**Authors:** Valentina Serra, Giancarlo Salvatori, Grazia Pastorelli

**Affiliations:** 1Department of Veterinary Medicine, University of Milano, Via dell’Università 6, 26900 Lodi, Italy; 2Department of Medicine and Science for Health “V. Tiberio”, University of Molise, Via Francesco De Sanctis 1, 86100 Campobasso, Italy; salvator@unimol.it

**Keywords:** cereals, *Zea mays*, *Hordeum vulgare*, *Triticum aestivum*, mycotoxins, beauvericin, enniatin B, oxidative stress, polyphenols, total antioxidant capacity

## Abstract

Increasing consumption of cereals has been associated with reduced risk of several chronic diseases, as they contain phytochemicals that combat oxidative stress. Cereal contamination by the “emerging mycotoxins” beauvericin (BEA) and enniatins (ENs) is a worldwide health problem that has not yet received adequate scientific attention. Their presence in feeds represents a risk for animals and a potential risk for humans because of their carry-over to animal-derived products. This preliminary study aimed to investigate if the total antioxidant capacity (TAC) of corn, barley, and wheat flours could be influenced by contamination with increasing levels of BEA and ENN B. The highest TAC value was observed in barley compared with wheat and corn (*p* < 0.001) before and after contamination. No effect of mycotoxin or mycotoxin level was found, whereas cereal x mycotoxin exhibited a significant effect (*p* < 0.001), showing a lower TAC value in wheat contaminated by ENN B and in barley contaminated by BEA. In conclusion, barley is confirmed as a source of natural antioxidants with antiradical potentials. Additional studies with a larger sample size are necessary to confirm the obtained results, and investigations of the toxic effects of these emergent mycotoxins on animals and humans should be deepened.

## 1. Introduction

Cereals are considered the most important sources of food [1] and are often classed as carbohydrate-rich foods, as they are composed of approximately 75% carbohydrate, among which starch is the main component. The main content in protein is about 6–15%; in wheat, the major storage proteins are represented by gliadins and glutenins; in corn, it is prolamin; and barley contains hordeins and glutelins [2]. Concerning lipids, these represent only a minor component of cereals, with the amount varying from 1–3% in barley, rice, and wheat, to 5–9% in corn, on a dry-matter basis. Cereals are an important source of most B vitamins, especially thiamin, riboflavin and niacin, and vitamin E. In the European Union (EU), wheat (*Triticum aestivum* L.) represents the most widely grown cereal crop, while the remaining 50% is composed of corn (*Zea mays*) and barley (*Hordeum vulgare),* each representing about one-third [3]. About two-thirds of cereal harvests within the EU are used for animal feed, while one-third are directed toward human consumption. Cereals and cereal products may also contain a range of bioactive substances, and there is growing interest in the potential health benefits these substances may provide. Among these bioactive compounds, phenolic acids, which are derivates of benzoic and cinnamic acids, represent the main phenolic compounds. A high consumption of cereals has been associated with a reduced risk of developing several chronic diseases [4], as they contain phytochemicals that combat oxidative stress in the body by helping to maintain a balance between oxidants and antioxidants. Phenolic compounds may provide health benefits associated with a reduced risk of chronic diseases such as anti-allergenic, anti-atherogenic, anti-inflammatory, antimicrobial, antioxidant, anti-thrombotic, cardioprotective, and vasodilatory effects. It is indeed well known that oxidative damage increases the risk of degenerative diseases such as cancer and cardiovascular pathologies [5,6].

Cereal grains are vulnerable to infections by a wide variety of plant pathogens, among which filamentous fungi show a remarkable potential to produce secondary metabolites such as mycotoxins [7]. In addition to the most studied and legislated mycotoxins, such as aflatoxins, ochratoxin A, fumonisins, and zearalenone, the so-called “emerging” mycotoxins enniatins (ENs) and beauvericin (BEA) are causing some concern about the risk and food safety [8]. BEA derives its name from the insect pathogenic fungus *Beauverina bassiana* from which it was first isolated [9]. It has been demonstrated that BEA decreases mitochondrial membrane potential, produces lipid peroxidation, DNA damage and cell death, interferes with smooth muscle contraction, and its toxicity is closely related to the dose, length, and way of exposure [10,11,12,13]. Moreover, its oxidative capacity has been demonstrated in studies conducted both in plants [14] and animals [11,15,16]. ENs represent a large group of emerging mycotoxins with the structure of cyclic hexadepsipeptides, comprised of D-α-hydroxy-isovaleryl-(2-hydroxy-3-methylbutanoic acid) and N-methylamino acid residues linked with peptide bonds and intra-molecular ester (lactone) bonds [17]. To date, 29 species of ENs have been characterized and, among them, ENs A, A1, B, and B1 are reported as natural contaminants in cereals in Europe [7]. ENs’ toxicity has been associated with the intracellular production of ROS and lipid peroxidation as observed in mammalian cells [11,18], although conflicting data on this aspect are present [19]. Several studies have reported the occurrence of BEA and ENs in livestock feedstuff. These mycotoxins have been described as frequent contaminants in grains, 50–90% for wheat, corn, and barley, contaminants in grains, potatoes, and apples used as food and livestock animals feed in several countries [8,20,21,22,23,24]. However, these so-called emerging mycotoxins have not yet received adequate scientific attention. To date, limits for BEA and ENs concentration have not been established, although their presence has been assessed by EFSA in feed at high levels (up to mg/kg or ppm) [25]. Furthermore, the determination of total antioxidant capacity (TAC) has gained growing interest as a tool for exploring the putative role of antioxidant-rich products in the prevention of degenerative diseases [26], to evaluate possible benefits from an antioxidant-enriched diet, or to verify if animals are experiencing oxidative stress [27].

For the aforementioned reasons, this pilot study aimed to investigate if the total antioxidant capacity of three cereals commonly used in animal feed could be influenced by contamination with increasing levels of BEA and ENN B mycotoxins.

## 2. Results and Discussion

In this pilot study, barley, wheat, and corn were characterized with the focus on total phenolic content (TPC) and antioxidant capacity (TAC) in order to investigate whether contamination with ENN B and BEA mycotoxins could exert an effect on it. The preliminary test on ENN B mycotoxin showed that the highest level of contamination produced values comparable to the control sample; thus, this finding suggests skipping the low concentration. Prior to determining the TPC and TAC values, the mycotoxin content was determined. Results showed that none of the samples was contaminated by BEA, ENN A, A1, B, and B1 since no detectable (nd) levels for all mycotoxins were found, with the exception of corn flour, which presented a level < 50 μg/kg. The limit of quantification was 50 μg/kg, in agreement with the Directive 2002/32 [28] and subsequent amendments on undesirable substances in animal feed. The Folin–Ciocalteu (F-C) assay is widely used to evaluate the in vitro TPC of cereal extracts. In the present pilot study, the TPC found in wheat, corn, and barley was 18.15 ± 1.30 vs. 19.89 ± 1.46 and 32.51 ± 3.04 mg TAE/100 g, respectively. Our results match the main findings of several studies demonstrating that these compounds represent a significant fraction in cereals [29,30] as much as other food groups, i.e., fruits, vegetables [31,32]. Durazzo et al. [30] reported, when studying total polyphenols in Italian soft wheat, that TPC values varied from 165.57 to 183.75 mg/100 g in aqueous-organic extracts in grains, while whole flours exhibited a more significant reduction than grain. In the study conducted by Horvat et al. [3], the highest TPC was found in barley (1322–1448 μg GAE/g) and the lowest in wheat (713–1032 μg GAE/g). Niroula et al. (2019) [33] found in corn the lowest TPC content (985 μg/g) in comparison to wheat (1091 μg/g) and barley (1274 μg/g). Figure 1 presents the results concerning the TAC values of the three cereal flours analyzed before mycotoxin contamination (negative controls). The highest TAC value (*p* < 0.001), expressed as mM uric acid equivalents, was observed in barley compared with wheat and corn, according to the highest phenolic content found that likely produced the stronger antioxidant capacity and an assortment of other bioactive phytochemicals [34].

These values are in agreement with those presented by Yamanouchi et al. [35], in which the mean value of antioxidant capacity of various plant samples was equal to 0.646 mM uric acid equivalents. This result is in agreement with several studies that demonstrated the higher antioxidant capacity of barley compared with other cereals [3,36,37]. In the study conducted by Madhujith and Shahidi [38], the higher antioxidant capacity of barley compared with that of oats and wheat was attributed to the total polyphenol content, which was higher in barley (32.4 mg gallic acid equivalent; GAE) than in wheat (9.2 mg GAE/g) and oats (27 mg GAE/g). In addition, in the study conducted by Pellegrini et al. [39], in which the determination of the antioxidant capacity was measured with the TEAC test (Trolox Equivalent Antioxidant Capacity), a higher value was found in barley equal to 2.34 mmol Trolox compared with corn, which presented a value equal to 0.65 mmol Trolox/kg. The strong positive relationship between the TAC and the TPC results found in mentioned papers and the consistency of our results suggests that the contribution of TPC to the TAC might be high. Even considering only the contaminated samples, barley showed the highest values (*p* < 0.001) compared with corn and wheat (0.98 vs. 0.54 and 0.44 mM uric acid equivalents). There was no significant effect of either mycotoxin or mycotoxin level, whereas cereal x mycotoxin exhibited a significant effect (*p* < 0.001). The antioxidant capacity of cereals did not seem to vary even after contamination with BEA and ENN B, showing similar values of their respective negative controls; we speculate that the phenolic compound content exerts a protective effect toward contamination. Similarly, Seo et al. [36] reported that the highest antioxidant properties found in barley extract were reflected in the inhibition of UV-induced lipid peroxidation performed in vitro. Both mycotoxins and UV are stressors inducing oxidative stress.

Phenolic acids are toxic to many fungi, including *Fusarium* species. In the last years, several studies support the contribution of grain antioxidant secondary metabolites to the mechanisms of plant resistance to *Fusarium* and mycotoxins accumulation. In fact, several studies have reported that antioxidant secondary metabolites present in cereals can modulate the production of mycotoxins by various fungal pathogens in addition to exerting antifungal properties [40]. The contamination with ENN B produced a reduction (*p* < 0.05) in the wheat TAC value compared with the BEA contamination. The negative influence of ENN B contamination was not observed in corn and barley flours, in which this value was significantly higher (*p* < 0.05; Figure 2). The involvement of ENs mycotoxins in initiating oxidative stress mechanisms has been demonstrated in a recent study conducted on Wistar rats, in which the exposure to concentrations of ENs A, A1, B, and B1 decreased the activity of the antioxidant enzyme superoxide dismutase 1 as a consequence of oxidative stress [41]. We speculate that this effect could be transposed to a different matrix, wheat in our case, although the extrapolation of in vitro data to our context is not straightforward and should hence be performed with caution. On the other hand, it is known that, at least in the case of wheat, exogenously applied enniatins can cause tissue necrosis that has been demonstrated to be correlated to oxidative stress and, consequently, to a reduced antioxidant capacity [42]. The result found in barley could be due to the highest content of polyphenols and probably to a defense mechanism that seems to inhibit the progression of pathogenic organisms such as fungi [43]. In fact, this property led us to hypothesize that this mechanism could have contributed to reducing the effect of mycotoxins contamination on the antioxidant capacity of this cereal.

ENN B is currently the most often detected in unprocessed and processed grains from European countries [20,44,45]. The carry-over of ENs and BEA from feed to animal-derived products is possible, as traces of these mycotoxins have been found in laying hens’ eggs, in some tissues of broilers and turkeys [46], and in milk [47]. This possibility could pose a health risk for consumers and makes further investigation necessary, especially because legal maximum levels and tolerable daily intake have yet to be established for these mycotoxins in any type of food commodity, probably because of limited data available about their toxicity, concentration levels, and occurrence [48].

In conclusion, barley is confirmed as an excellent dietary source of natural antioxidants with antiradical potentials. Moreover, the contamination under the present experimental conditions did not produce a reduction in antioxidant power in this cereal. If the results of contaminated wheat with ENN B found in this pilot study will be confirmed, it could represent a crucial point and makes further investigation necessary. It should be noted that, for the preparation of animal feed, lower quality grains are used that are prone to mold contamination and the presence of mycotoxins. Given the possible carry-over of ENs and BEA from feed to animal-derived products, additional studies are still necessary to investigate their distribution and contamination levels as well as their toxic effects on humans and animals.

## 3. Materials and Methods

### 3.1. Materials and Reagents

Flour samples of corn (*Zea mays* L.), barley (*Hordeum vulgare* L.), and wheat (*Triticum aestivum* L.) were supplied by a local feed mill located in North Italy. All samples were packed in polyethylene bags and stored at −20 °C in the dark until analyses. Chemical materials used in this study were of HPLC-grade and were purchased from Sigma-Aldrich Canada Ltd. (Oakville, ON, Canada): beauvericin (CAS No. 26048-05-5), enniatin B (CAS No. 917-13-5), methanol (CAS No. 67-56-1), dimethyl sulfoxide (DMSO; CAS No. 67-68-5), acetone (CAS No. 67-64-1), tannic acid (CAS No. 1401-55-4), Folin–Ciocalteu reagent (cat. no. F9252), and sodium carbonate (CAS No. 497-19-8). Water for analyses was purified by the Milli-Q system (Millipore Corp., Milford, MA, USA). Stock solution of BEA (1 mg/mL) was prepared in methanol, while stock solution of ENN B (10 mg/mL) was prepared in DMSO. Both stock solutions were then diluted to obtain the appropriate work solutions and were maintained at −20 °C until use.

### 3.2. Determination of Mycotoxins

The determination of BEA and ENs (ENN A, A1, B, B1) content was performed by a French commercial laboratory (Phytocontrol Agrifood, Nîmes, France). This institution is accredited by the Council for Accreditation, Audit, and Control (COFRAC) according to LAB GTA 21 (Program 99-1) for the determination of mycotoxins in all foodstuffs intended for humans, animals, and infant food [49]. In accordance with this standard, an internal liquid chromatography-tandem mass spectrometry (LC-MS/MS) method was used.

### 3.3. Total Phenolic Content Assay

The total phenolic content (TPC) in cereal extracts was determined with Folin–Ciocalteu (FC) reagent according to Attard (2013) [50]. Prior to performing the assay, 5 g of each cereal flour sample was allowed to macerate for 72 h with 20 mL methanol in Falcon tubes, mixing samples once daily with a vortex for 2 min. After centrifugation (Hermle Z326K, Wehingen, Germany) at 1500 rpm and 4 °C for 10 min, the supernatant was recovered and stored in the dark at −20 °C until analysis. Tannic acid standard solution (960 μg/mL) was serially diluted 1:2 up to 60 μg/mL to construct the calibration curve. The FC reagent was diluted 1:10 with deionized water, while sodium carbonate was prepared as a 1 M solution. Briefly, 10 μL of cereal extracts or tannic acid dilutions were pipetted in triplicate in wells of a 96-well microtiter plate (Nunc™ MicroWell™, ThermoFisher Scientific™). After adding 100 μL of FC reagent and 80 μL of sodium carbonate to each well, the plate was allowed to incubate at room temperature in the dark for 20 min and read at 630 nm on a spectrophotometer (SpectraMAX 340PC, Molecular Devices Corporation, San Jose, CA, USA). The total phenolic content was expressed as mg tannic acid equivalents (TAE)/100 g.

### 3.4. Extraction Procedure for Total Antioxidant Capacity Assay

#### 3.4.1. Sample Treatment

##### Negative Controls

Prior to performing the total antioxidant capacity (TAC) assay, the flour samples were subjected to an extraction phase. Flour (1 g of each sample: maize, barley, and wheat) was weighed and placed in Falcon tubes. Each sample was homogenized for 5 min using a vortex (ArgoLab, Vortex Mixer) after adding 50% methanol (1:2, *w*/*v*). After centrifugation at 10,000× *g* for 10 min at 4 °C, the supernatant (the water-soluble fraction) was recovered and transferred in Eppendorf tubes and stored at −20 °C until analysis. The insoluble fraction (pulp) of each sample was further extracted by adding pure acetone (1:4, *w*/*v*) and mixing at room temperature for 30–60 min using an orbital shaker incubator. After the incubation time, the pulp was centrifuged at 10,000× *g* for 10 min at 4 °C, and the supernatant was recovered and diluted with PBS prior to running the assay. In this study, both the supernatants were analyzed, and the TAC values were calculated by combining the results from the water-soluble fraction and the acetone extract from the pulp fraction.

##### Contaminated Samples

Mycotoxin concentrations were chosen to simulate a realistic scenario. Flour samples were exposed for 24 h to BEA (0.5 mg/kg, 3 mg/kg, and 5 mg/kg) and ENN B (2.5 mg/kg and 5 mg/kg) at room temperature using an orbital shaker incubator to ensure mixing [51]. Subsequently, each sample was subjected to extraction steps as previously described.

### 3.5. Total Antioxidant Capacity (TAC) Assay

The total antioxidant capacity (TAC) of flour samples was measured using the commercial kit OxiSelect™ Total Antioxidant Capacity Assay (Cell Biolabs, Inc., San Diego, CA, USA) according to the manufacturer’s instructions. This assay is based on the reduction of Cu^2+^ to Cu^+^ by endogenous antioxidants. Upon reduction, the Cu^+^ further reacts with a coupling chromogenic reagent that produces a color with a maximum absorbance at 490 nm. Each sample was dispensed in a 96-well microtiter plate (Nunc™ MicroWell™, ThermoFisher Scientific™), after which 180 μL of 1X reaction buffer was added to each well and mixed. An initial reading at 490 nm was taken for each sample. Then, 50 μL of 1X copper ion reagent was added and incubated for 5 min on an orbital shaker. Next, 50 μL of stop solution was added to terminate the reaction, and the plate was subjected to additional measurement of absorption at 490 nm using the spectrophotometer SpectraMAX 340PC (Molecular Devices Corporation). All determinations were performed in duplicate, and results were averaged. TAC of each sample was determined using a calibration curve based on known concentrations of uric acid standards. The results were expressed as mM uric acid equivalents. In this study, the TAC values were calculated by combining the results from the water-soluble fraction and the acetone extract from the pulp fraction.

### 3.6. Statistical Analyses

Statistical analyses were performed with SPSS software (SPSS/PC Statistics 26.0, SPSS Inc., Chicago, IL, USA). The total antioxidant capacity results were analyzed using univariate analysis, with cereal, mycotoxin, and the level of mycotoxin contamination as fixed factors. Comparisons between means were evaluated with the Newman–Keuls test. Differences were deemed significant at *p* < 0.05, and a trend was noted when *p* < 0.1.

## Figures and Tables

**Figure 1 plants-10-01835-f001:**
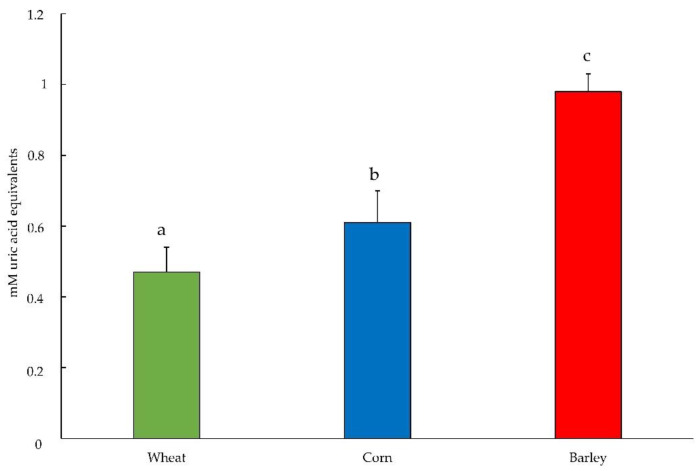
Total antioxidant capacities (in mM uric acid equivalents) of cereal flours (negative controls) measured by OxiSelect™ Total Antioxidant Capacity Assay (*p* < 0.001). Different letters above the bars indicate significant differences at *p* ˂ 0.05.

**Figure 2 plants-10-01835-f002:**
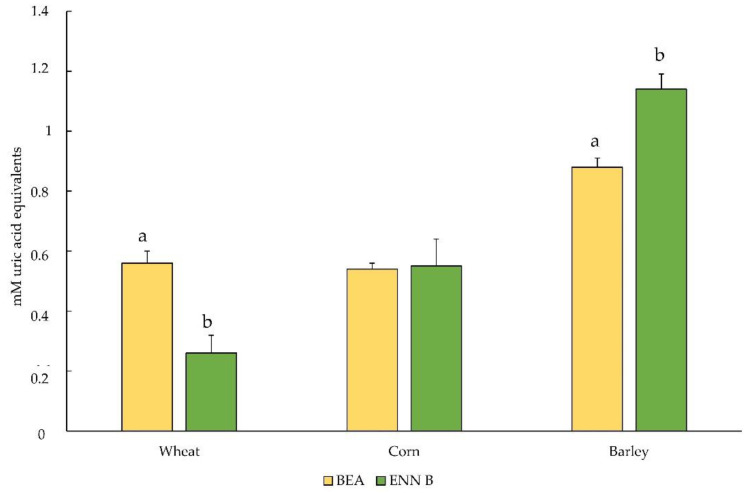
Total antioxidant capacity values (in mM uric acid equivalents) of cereal flours contaminated with BEA and ENN B mycotoxins: BEA, beauvericin; ENN B, enniatin B. Different letters above the bars indicate significant differences at *p* ˂ 0.05.

## Data Availability

The data presented in the current study are available from the corresponding authors on reasonable request.

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
