# Peer review of "Pilot Study: Does Contamination with Enniatin B and Beauvericin Affect the Antioxidant Capacity of Cereals Commonly Used in Animal Feeding?"

_plants, 2021, doi:10.3390/plants10091835_

Round 1

Reviewer 1 Report

This research article highlights the importance of analysing commonly used cereals contaminated and uncontaminated with “emerging mycotoxins” beauvericin and enniatins.

Considering that wheat, corn and barley are the most widely grown cereal crop in Europe, the topic is of great interest.

Therefore, this study aimed to analyse the total antioxidant activity of three cereal crops with different mycotoxin contamination levels.  

The article is well written, however, using CUPRAC assay for total antioxidant capacity (TAC) can provide limited information about the antioxidant status. CUPRAC assay does not measure all antioxidant components, indicating that different results can be obtained when different assays are used to measure TAC like: DPPH, ABTS, FRAP, TEAC.

I strongly recommend using at least one other complementary method for analyzing the total antioxidant capacity.

Author Response

Response to Reviewer 1 Comments 

This research article highlights the importance of analysing commonly used cereals contaminated and uncontaminated with “emerging mycotoxins” beauvericin and enniatins. 

Considering that wheat, corn and barley are the most widely grown cereal crop in Europe, the topic is of great interest. 

Therefore, this study aimed to analyse the total antioxidant activity of three cereal crops with different mycotoxin contamination levels. 

The article is well written, however, using CUPRAC assay for total antioxidant capacity (TAC) can provide limited information about the antioxidant status. CUPRAC assay does not measure all antioxidant components, indicating that different results can be obtained when different assays are used to measure TAC like: DPPH, ABTS, FRAP, TEAC. 

I strongly recommend using at least one other complementary method for analyzing the total antioxidant capacity. 

Thanks for your effort and time spent on our manuscript. 

We used CUPRAC assay since this method presents advantages like simplicity, availability and stability of reagents, reproducibility over a wide concentration range, selection of working pH at physiological pH, applicability to both hydrophilic and lipophilic antioxidants (Apak et al., 2007, “Comparative evaluation of various total antioxidant capacity assays applied to phenolic compounds with the CUPRAC assay”; Özyürek et al., 2011, “A comprehensive review of CUPRAC methodology”). 

The same assay kit (OxiSelect Total Antioxidant Capacity Assay Kit; CELL BIOLABS, INC., USA) has been successfully used to determine the phytochemicals content of some sprout vegetables (Yamanouchi et al. (2018) 

Reviewer 2 Report

The study is original and innovative in the field. The study have a high level of novelty, being made on two new mycotoxins ( beauvericin  and enniatins) whose effect is not yet well studied in international literature. The authors have  investigated  how  the total antioxidant capacity  of corn, barley and wheat flours could be influenced by contamination with these mycotoxins.
The leve of English language is very good.

Author Response

Response to Reviewer 2 Comments 

The study is original and innovative in the field. The study have a high level of novelty, being made on two new mycotoxins (beauvericin and enniatins) whose effect is not yet well studied in international literature. The authors have investigated how the total antioxidant capacity of corn, barley and wheat flours could be influenced by contamination with these mycotoxins. The level of English language is very good. 

We thank the positive appraisal from reviewer on our work and the manuscript itself. 

Reviewer 3 Report

The manuscript “does contamination with enniatin B and beauvericin affect the antioxidant capacity of cereals commonly used in animal feeding”The findings are quite preliminary and also the presentation needs to improve. Besides, few points need to address. 

  1. The number of samples is too low so focus on the samples size.
  2. Also, the author focused only on the phenolics for the antioxidant. There are other phytochemicals that also have a great impact on antioxidant activity.
  3. In line 20 what kind of negative and positive influence is exerted by ENN B?? Mention
  4. The introduction part needs to improve particularly the first paragraph “Cereals…. at human consumption” is irrelevant.
  5. Mention the used LC-MS conditions in the material section.
  6. Inline no 50-51 remove anti before anti-allergenic, anti-atherogenic, anti-inflammatory, anti-microbial, antioxidant, anti-thrombotic, cardioprotective, and vasodilatory effects.
  7. In line 197, 5mg/ml rewrite as 1 mg/ml
  8. Line 213, Tannic acid was prepared in five 1 in 2 dilutions. What does it mean? Make it clear.
  9. Address the grammatical and typo mistake throughout the MS e.g μl or μL (line 250)

Author Response

Response to Reviewer 3 Comments 

The manuscript “does contamination with enniatin B and beauvericin affect the antioxidant capacity of cereals commonly used in animal feeding”. The findings are quite preliminary and also the presentation needs to improve. Besides, few points need to address. 

We are grateful for your valuable comments and suggestions, which were very useful to improve our paper. We have carefully revised the manuscript and highlighted all changes. Please find our response on the questions asked below. 

1. The number of samples is too low so focus on the samples size 

We added the term “pilot study” (see lines 85, 96 and 179). We are aware that samples considered were few; anyway, we underlined that the experiment is a pilot study and therefore aims to examine the feasibility on a small scale of the approach that is intended to be used in a larger scale study. 

2. Also, the author focused only on the phenolics for the antioxidant. There are other phytochemicals that also have a great impact on antioxidant activity. 

We agree with the comment of reviewer and modified the sentence adding a new reference (number 34): “The highest TAC value (p < 0.001) expressed as mM uric acid equivalents, was observed in barley compared to wheat and corn according to the highest phenolic content found that likely produced the stronger antioxidant capacity and to an assortment of other bioactive phytochemicals (Idehen et al., 2017)”. 

3. In line 20 what kind of negative and positive influence is exerted by ENN B?? Mention 

We modified the sentence as follows: “No effect of mycotoxin or level was found, whereas cereal x mycotoxin exhibits a significant effect (p < 0.001) showing a lower TAC value in wheat contaminated by ENN B and in barley contaminated by BEA.” Lines 19-21. 

4. The introduction part needs to improve particularly the first paragraph “Cereals…. at human consumption” is irrelevant. 

Thank you for the suggestion, we deleted the first paragraph of the introduction. Lines 29-31. 

5. Mention the used LC-MS conditions in the material section. 

The protocol used in LC-MS/MS is an internal method of the accredited laboratory Phytocontrol Agrifood (France). We modified the sentence as follows: “The determination of BEA and ENs (ENN A, A1, B, B1) content was performed by a French commercial laboratory (Phytocontrol Agrifood, Nîmes, France). This institution is accredited by Council for Accreditation, Audit, and Control (COFRAC) according to LAB GTA 21 (Program 99-1) for the determination of mycotoxins in all foodstuffs intended for humans and animals, and infant food [49]. In accordance with this standard, an internal liquid chromatography-tandem mass spectrometry (LC-MS/MS) method was used.” Lines 201-207”, 

Since the analysis was performed by an external laboratory, we refer to the annex indicates in literature. 

6. Inline no 50-51 remove anti before anti-allergenic, anti-atherogenic, anti-inflammatory, anti-microbial, antioxidant, anti-thrombotic, cardioprotective, and vasodilatory effects. 

Done as suggested. 

7. In line 197, 5mg/ml rewrite as 1 mg/ml 

Done as suggested. Line 196. 

8. Line 213, Tannic acid was prepared in five 1 in 2 dilutions. What does it mean? Make it clear. 

We modified the sentence as follows: “Tannic acid standard solution (960 μg/ml) was serially diluted 1:2 up to 60 μg/ml to construct the calibration curve.” 

9. Address the grammatical and typo mistake throughout the MS e.g μl or μL (line 250) 

Thanks for your attention. The whole manuscript has been checked to avoid any other such mistakes. 

Round 2

Reviewer 1 Report

From lines 49 to 53, if you delete "anti", you change the meaning of the whole paragraph. Please rewrite this accordingly. For example: "with reduced risk of chronic diseases such as allergies, atherogenesis, inflammation", and so on. Or, consider leaving it the way it was. 

What is the utility for the first graph? Isn't the same information presented in the second graph?

For the mycotoxins determination, there are no results presented from the third-party laboratory that performed the analysis. Are there any data available?

From lines 253 to 256 you changed μL to μl. Please reconsider using capital L, in order to avoid the risk of confusion between the letter l (el) and the numeral 1 (one).

Author Response

Dear Reviewer, many thanks for your effort and time spent to improve the quality of our manuscript.

We have followed your recommendations and answered each of your questions.

From lines 49 to 53, if you delete "anti", you change the meaning of the whole paragraph. Please rewrite this accordingly. For example: "with reduced risk of chronic diseases such as allergies, atherogenesis, inflammation", and so on. Or, consider leaving it the way it was. 

Thanks for your attention. We agree with the reviewer’s comment and we modified the paragraph leaving it in the way it was. Lines 48-49.

What is the utility for the first graph? Isn't the same information presented in the second graph?

We deleted Figure 2 as the TAC values of contaminated samples are reported in the text (line 129). Accordingly, Figure 3 has now been renamed Figure 2.

For the mycotoxins determination, there are no results presented from the third-party laboratory that performed the analysis. Are there any data available?

The results concerning mycotoxins determination are reported in Lines 90-93: “Results showed that none of the samples was contaminated by BEA, ENN A, A1, B and B1 since no detectable (nd) levels for all mycotoxins, with exception of corn flour that presented a level < 50 μg/kg, were found.” The report of analyses performed by Phytocontrol laboratory showed the results expressed as “nd” (not detect), since values were under the limit of detection (50 μg/kg).

From lines 253 to 256 you changed μL to μl. Please reconsider using capital L, in order to avoid the risk of confusion between the letter l (el) and the numeral 1 (one).

Thanks for the suggestion. We checked throughout the manuscript and corrected according to your comment using capital L.

Reviewer 3 Report

Suggested changes have been incorporated in the revised MS. I recommend to publish the MS.

Author Response

Suggested changes have been incorporated in the revised MS. I recommend to publish the MS.

Thanks for the time spent on our manuscript. We are grateful for your valuable comment and for giving us the opportunity to improve our work.

Round 3

Reviewer 1 Report

Suggested changes have been incorporated in the revised MS. I recommend publishing the MS.